# Experimental evaluation of stiffening effect induced by UVA/Riboflavin corneal cross-linking using intact porcine eye globes

Shao-Hsuan Chang[1,2☯¤a]*, Dong Zhou[1], Ashkan Eliasy[1], Yi-Chen Li[2☯¤b]*, Ahmed Elsheikh[1]

1 School of Engineering, University of Liverpool, Liverpool, United Kingdom, 2 Department of Chemical Engineering, Feng Chia University, Taichung, Taiwan

☯ These authors contributed equally to this work.
¤a Current address: School of Engineering, University of Liverpool, Liverpool, United Kingdom
¤b Current address: Department of Chemical Engineering, Feng Chia University, Taichung, Taiwan
* yicli@fcu.edu.tw (LYC); changshao1311@gmail.com (CSH)

**Data Availability Statement:** All relevant data are available from Dryad (doi: 10.5061/dryad. z8w9ghx9f).

## Abstract

UVA/riboflavin corneal cross-linking (CXL) is a common used approach to treat progressive keratoconus. This study aims to investigate the alteration of corneal stiffness following CXL by mimicking the inflation of the eye under the *in vivo* loading conditions. Seven paired porcine eye globes were involved in the inflation test to examine the corneal behaviour. Cornea-only model was constructed using the finite element method, without considering the deformation contribution from sclera and limbus. Inverse analysis was conducted to calibrate the non-linear material behaviours in order to reproduce the inflation test. The corneal stress and strain values were then extracted from the finite element models and tangent modulus was calculated under stress level at 0.03 MPa. UVA/riboflavin cross-linked corneas displayed a significant increase in the material stiffness. At the IOP of 27.25 mmHg, the average displacements of corneal apex were 307 ± 65 μm and 437 ± 63 μm (p = 0.02) in CXL and PBS corneas, respectively. Comparisons performed on tangent modulus ratios at a stress of 0.03 MPa, the tangent modulus measured in the corneas treated with the CXL was 2.48 ± 0.69, with a 43±24% increase comparing to its PBS control. The data supported that corneal material properties can be well-described using this inflation methods following CXL. The inflation test is valuable for investigating the mechanical response of the intact human cornea within physiological IOP ranges, providing benchmarks against which the numerical developments can be translated to clinic.

## 1. Introduction

Research in corneal cross-linking (CXL) has gradually developed a tool that became the first-line treatment for treating keratoconus. The prevalence of this disease ranges from 1 in 375 in the general population [1]). This technique creates new covalent cross-links between molecules and extracellular matrix within stroma to strengthen and stabilize the structure of cornea.

**Funding:** This work was supported by grants from the Tawian Ministry of Science and Technology (MOST 109-2221-E-035 -036 -MY3 and  MOST 109-2221-E-035 -007 -MY3).

**Competing interests:** The authors have declared that no competing interests exist.

Collagen cross-linking using the conventional Dresden protocol is initially thought to affect the mechanical properties, ultrastructure, hydrodynamic and enzymatic behavior of the cornea [2–7]. To assess the effectiveness of this treatment, biomechanical characterization provides quantitative measurements to assess the degree of cross-linking. Many studies reported a significant increase in the corneal stiffness through experimental studies on cross-linked corneas measured by tensile or inflation tests [6,8–10]. However, various range of stiffness is proposed due to the variations in methodologies used for experimental settings. In addition, the estimated tangent modulus using tensile test is considerably higher than the measurement from inflation, which can vary by the order of magnitude. Therefore, in this study, a unique inflation test rig was designed and developed which allowed characterisation of intact porcine eye globes for more accurate measurement and predictability of CXL.

The stiffness of the cornea was determined by its nonlinear geometry and material behavior. The material behavior was highly related to its micro- and ultra-structure, including the hydration, spatial fibril density and arrangement [11]. Uniaxial tensile test is the most commonly used technique for the measurement of corneal stiffness. However, the mechanical measurements may be inaccurate due to its technique destructive specimen preparation with disrupted fibril orientations, inconsideration of corneal curvature, and non-uniform stress distribution while applied the stretching force along the corneal strip [12–14]. Therefore, inflation test has been developed and attempted to address this issue by mimicking the *in vivo* loading conditions of the eye, which is considered to be more reliable and closely related to *in vivo* conditions than uniaxial tensile test. It is expected to produce the average behavior of intact corneas due to the stromal anisotropy resulting from the preferred collagen fibril orientation. Inflation test analyses the degree of extension of the cornea in response to the change in IOP.

During the test, a variety of monitoring techniques including the use of laser or DIC can be applied to track displacements [15–17]. For post-test analysis, the finite element (FE) modelling technique is employed to construct numerical models of whole corneas or eye globes, and the material stress-strain relationships can be adjusted until the predicted surface deformations of the models matched those observed experimentally. This technique has provided an accurate means of determining the tissue's tangent modulus [18,19].

Due to the non-uniform curvature with variable thickness, both the external and internal geometries of eye globe were obtained to generate the corneal model. A model developed for the cornea using entire eye globe allowed appropriate realistic displacement at the limbus, the displacements at the limbus were tracked during the experiment and then introduced at the boundary of the corneal FE model. This approach of performing a cornel only model was to establish a biomechanical model that can mimic the functional response of a real eye with the reduced geometrical complexity and increased efficiency of the computational calculation. Paired eye globes were used to examine the intact corneal behaviors, with one as the test sample with CXL treatment and the other as its non-treated control in PBS. Through inverse analysis, non-linear material behaviours were better understood with the effect of Dresden cross-linking protocol on stiffness of the tissue. This study was also the first one to consider the stiffness of cornea as a whole following CXL. These results may potentially supervise the further modelling of CXL treatment in order for the accurate surgical prediction, which may allow clinicians to estimate the severity of keratoconus and accomplish the direct comparison with the refractive outcomes in practice.

## 2. Method

### 2.1. Specimen and preparation

Seven paired fresh porcine eyes were obtained from a local abattoir (Morphets, Tan house farm, Widnes) and tested within 6–9 hours after death. Soft muscular tissue was removed with

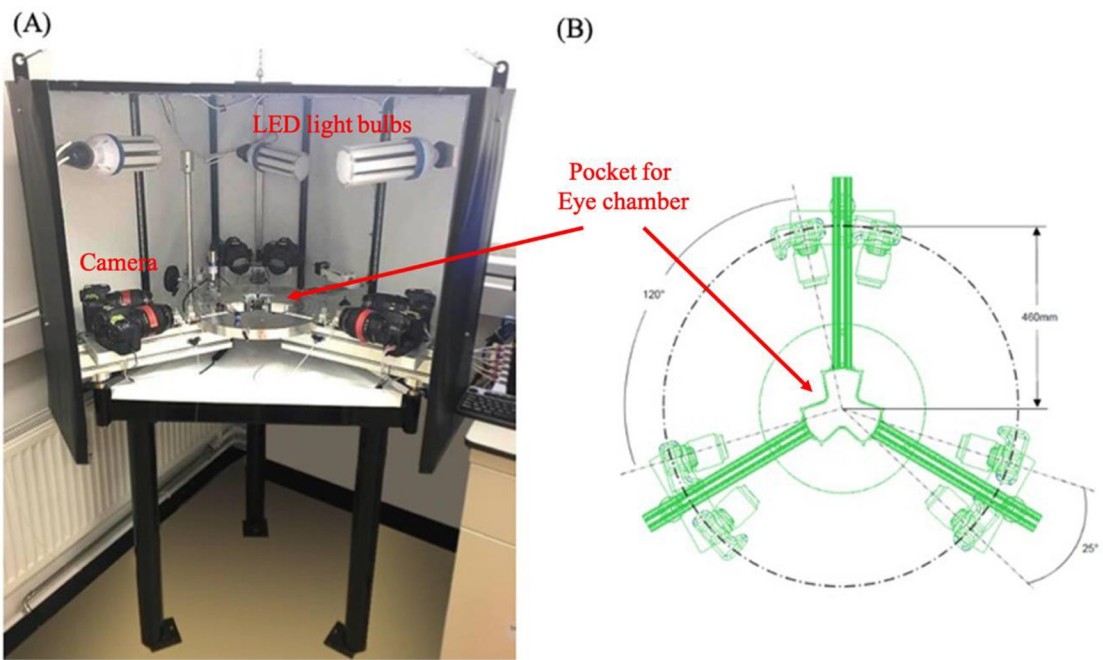

**Fig 1. Inflation test equipment.** (A) inflation setup with front cover removed, (B) camera array showing angles between cameras and distance from cameras to eye chamber (placed in the centre).

surgical scissors. The superior direction was marked and the eye globe was placed in a customized compartment for accurate needle insertion through the posterior pole. The internal eye components were removed through the posterior pole using a 14G needle. The needle was then lightly glued around the posterior pole and the intra-ocular cavity was washed with 5 to 6 ml PBS (Sigma, Dorset, United Kingdom). The outer surface of the globe was continually kept hydrated by applying PBS every 2–5 minutes. Random speckles were applied on the globe by lightly spraying a waterproof and fast drying black paint to facilitate deformation tracking in post-analysis. The prepared specimen was then placed into a custom-designed eye chamber filled with PBS, and transferred onto the inflation rig (Fig 1).

## 2.2. Dresden protocol

The right eyes of the pared specimens were prepared for CXL prior to inflation test. The procedure was performed following conventional Dresden protocol [20], with the anterior surface of the corneas applied with 5 mL of 0.1% riboflavin in dextran at 3-minute intervals treating for 30 minutes. UVA (370 nm) illumination at 3 mW/cm2 (Opto Xlink; Mehra Eyetech Pvt. Ltd., Delhi, India) was then performed for further 30 minutes. Topical dosing of riboflavin drops was continued during the irradiation.

## 2.3. Test rig

The inflation test rig provides full-field observation of ocular response to uniform intraocular pressure (IOP) changes. The physical test equipment is fully bespoke having been designed and built in-house (Fig 1). The equipment features closed loop control software written in LabVIEW (version 10.0.1, RRID:SCR_014325) to regulate IOP while collecting real-time data by triggering cameras to take pictures of the globe. The obtained images are used for measurement of deformation across the globe. The specimen was clamped in a horizontally placed eye

chamber with high precision real-time laser (LK-2001, Keyence, UK) pointing towards the apical displacement. An array of six high resolution digital cameras (18.0 megapixels, 550D, Canon, Tokyo, Japan) surrounding the eye chamber and a pressure adjusting tank was placed vertically to inflate the eye while taking synchronous images. The camera setup shown in Fig 1B allows an angle of 25˚ within each pair and an angle of 120˚ between each set.

## 2.4. Testing control and protocol

A custom-built LabVIEW software was used to tightly control the pressure. The experiments started by 3 pre-conditioning cycles. The pre-conditioning cycles were to ensure the eye was sitting comfortably on the needle, and the tissue behavior was repeatable [15] An initial pressure of 2.5 mmHg was used to balance the external pressure applied by PBS in the pressure chamber, and was therefore considered a zero-pressure point for the inflation test.

Specimens were loaded to a maximum internal load at a medium rate of 0.55 mmHg/s for each cycle. During each cycle the eye was allowed to relax for a period of 2 minutes which was obtained experimentally to allow tissue to fully recover to its relaxation state. The behavior of specimen in the final loading cycle was used for post-analysis.

## 2.5. Thickness measurement

After the experiment was completed, the eye was removed from the test rig and dissected into anterior and posterior parts. Eight meridian profiles of discrete thickness measurements were selected as shown in Fig 2. The thickness at each desired point on each meridian line was determined using an in-house developed Thickness Measurement Device (TMD) (LTA-HS, Newport, Oxfordshire, UK) which was developed by the Biomechanical Engineering group to measure the thickness of biological tissue. A vertical measurement probe was located at a height of about 30 mm above the centre point of the support. The probe moved down with a controlled velocity until it reached the surface of the tissue. By precisely knowing the original distance between the initial position of probe and the surface of support, the measured value was recorded as the thickness of the tissue.

## 2.6. Geometric modelling

To decrease the geometrical complexity and understand the effect of CXL treatment on corneas where the application of interest is, we built up a corneal-only model by excluding the sclera part from a whole globe model. In this corneal model, the orphan mesh of geometry was constructed with Abaqus 6.13 (Dassault Systèmes Simulia Corp., Rhode Island, USA) using bespoke software. The 2592 elements with 8611 total nodes adopted the hybrid and quadratic

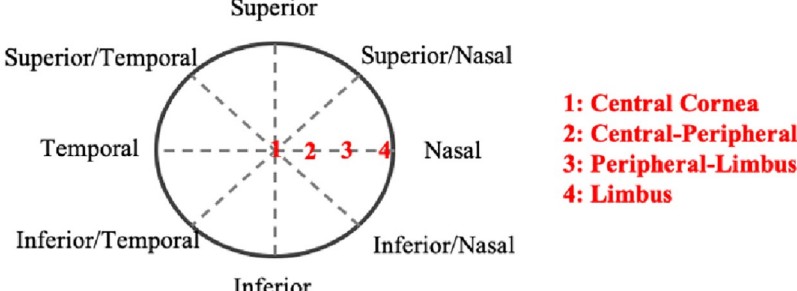

**Fig 2. Thickness measurement was performed on each sample along the eight meridian lines with four points measured per line.**

type with triangular cross-section (C3D15H), which were arranged in 12 rings across the cornea surface and 3 layers through the thickness. Corneal apex was restrained against displacement in X- and Y-directions, whereas limbus was restrained in the X-, Y-, and Z-direction. The intraocular pressure was distributed on the posterior surface of the cornea. The apical displacement of the entire cornea was extracted by the displacement of corneal apex minus the average displacement of limbus in the anterior-posterior direction.

## 2.7. Deformation measurement by Digital Image Correlation (DIC)

The image profiles obtained were analyzed using a 2D DIC method named Particle Image Velocimetry (PIV) to obtain deformations on the surface of the eye (Fig 3) [21,22]. PIV compares an un-deformed and deformed image pairs of specimen surface which was speckled to present the local displacements within the selected subsets. Three discrete locations including corneal apex and limbus were measured from each camera (Fig 3B). As only cornea was considered in the study, the cornea deformation was calculated by subtracting the average displacement of limbus in the anterior-posterior direction from the displacement of corneal apex.

## 2.8. Determining the corneal material properties

An in-house built software that uses Particle Swarm Optimization (PSO) as an optimization strategy was developed in Matlab (RRID:SCR_001622) to conduct the inverse analysis optimization due to its success in the engineering applications [23–25]. PSO evaluates the fitness of the apical displacement between simulation and experiment and iterates over the different values of material parameters to decrease the error until the best fitness appears. The material constitutive model chosen to demonstrate the material behavior of ocular tissue during loading was Ogden model as presented in Eq 2.1, utilized in a number of previous studies on soft tissues [26,27].

$$W(\lambda_1, \lambda_2, \lambda_3) = \sum_{i=1}^{N} \frac{2\mu_i}{\alpha_i^2} \left( \bar{\lambda}_1^{\alpha_i} + \bar{\lambda}_2^{\alpha_i} + \bar{\lambda}_3^{\alpha_i} - 3 \right) + \sum_{i=1}^{N} \frac{1}{D_i} (J-1)^{2i} \tag{2.1}$$

where $W$ is the strain energy density; $\bar{\lambda}_i$ are the deviatoric principal stretches, $\bar{\lambda}_i = J^{-\frac{1}{3}}\lambda_i$; $\lambda_i$ are the principal stretches; $J$ denotes the determinant of deformation gradient and describes the

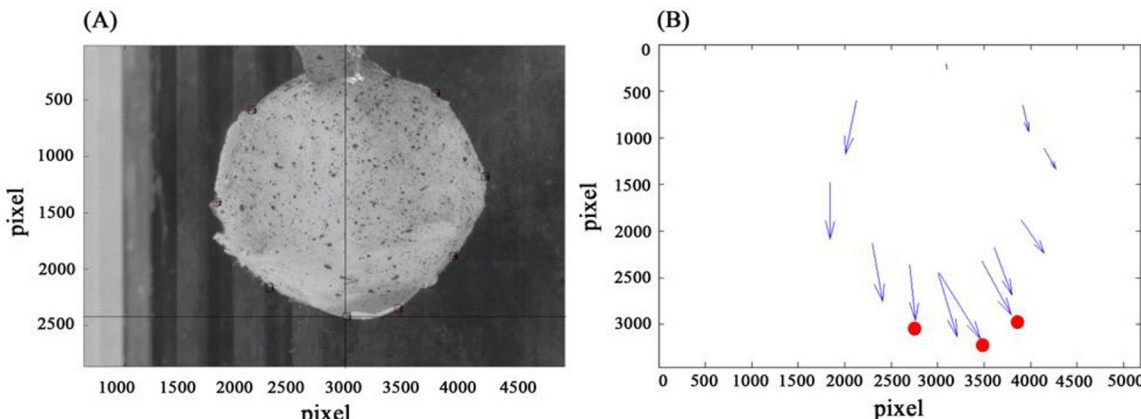

**Fig 3. A demonstration of Particle Image Velocimetry (PIV).** (A) screen capture of the manual tagging of desired points. (B) result of deformed tagged points (corneal apex and limbus). mm/pixel = 1.

change of material volume; the second term was ignored as the cornea tissue was incompressible ($J = 1$). $\alpha_i$ and $\mu_i$ are material parameters; N is the function order and N = 1 was used in this study.

The Ogden material model order one relies on two parameters of μ (shear modulus) and α (strain hardening exponent) to define the non-linear material behavior. The use of first order material model (N = 1) reduced the complexity of optimization and thus the computational cost as a result of less variables. The values of material parameters α and μ represented the output of the inverse modelling process that resulted in the highest fitness of simulation against inflation experiment. Therefore, the objective function was to minimize the root mean squared (RMS) of deformation, which was calculated as shown in Eq 2.2:

$$RMS\% = \frac{1}{M}\sum\nolimits_{j=1}^{M}\frac{\sqrt{\frac{1}{N}\sum_{i=1}^{N}\left(\delta_{i,j}^{experimental} - \delta_{i,j}^{numerical}\right)^{2}}}{\delta_{maxj}^{experimental}}X100 \tag{2.2}$$

where M is the number of measurement locations; N is the number of pressure levels; $\delta_{i,j}$ is the deformation at each particular pressure level $i$ and location $j$.

The design optimization process adjusts the value of μ and α within the constitutive model while setting a wide lower and upper boundary range (lower boundary = [0.005, 50]; upper boundary = [0.2, 200]). The error limit of RMS was set as 10%, which terminated the optimization once the error is lower than the limit. With these parameters, stress and strain could then be extracted from the numerical modelling results. The uniaxial-mode stress was calculated through obtained μ and α in Table 2, based on the previously described method [28] and then tangent modulus was calculated numerically from the gradient of the resulting stress-strain curve by Eq 2.3:

$$E = \frac{\Delta\sigma}{\Delta\varepsilon} \tag{2.3}$$

Where $\sigma$ is stress and $\varepsilon$ is strain. The strain difference $\Delta\varepsilon$ in this study is 0.2%, the corresponding stress values at each strain value were presented as shown in Fig 6.

## 2.9. Statistical analysis

The statistical evaluation was performed using SPSS software version 18.0 (IBM Corp. USA, RRID:SCR_002865). Results are expressed as means ± standard deviation (SD) and significant differences are calculated using one-way analysis of variance (ANOVA) with Turkey's HSD post-hoc test. Significance differences accepted where $p < 0.05$.

## 3. Results

By providing the pressures from posterior pole to the corneal apex with respect to IOP, the measurements of displacement indicate the response of the eye globes. The thickness variations across the whole cornea were demonstrated in Table 1. Four points per line across the

**Table 1. Average thickness measurements across the whole cornea at four different points.** The thickness was measured after the inflation test.

| Corneal regions | No. of Samples | Thickness of PBS Control Eye (mm) | Thickness of Cross-linked Eye (mm) | p value |
|---|---|---|---|---|
| Central cornea | 7 | 1.34 ± 0.09 | 0.95 ± 0.08 | 0.003 |
| Central-peripheral | 7 | 1.19 ± 0.05 | 1.03 ± 0.07 | 0.02 |
| Peripheral-limbus | 7 | 1.17 ± 0.06 | 1.03 ± 0.06 | 0.02 |
| Limbus | 7 | 1.14 ± 0.02 | 1.07 ± 0.05 | 0.06 |

**Table 2. Optimised material parameters α and μ obtained for all specimens from the corneal region using inverse modelling procedure.**

| Parameters | Control (left eye) | | CXL (right eye) | |
|:---:|:---:|:---:|:---:|:---:|
| | μ | μ | μ | μ |
| **EYE1** | 0.0102 | 96.07 | 0.0403 | 112.54 |
| **EYE2** | 0.0100 | 57.30 | 0.0139 | 73.31 |
| **EYE3** | 0.0091 | 64.69 | 0.0126 | 80.93 |
| **EYE4** | 0.0144 | 75.52 | 0.0112 | 151.22 |
| **EYE5** | 0.0082 | 57.65 | 0.0088 | 90.42 |
| **EYE6** | 0.0096 | 53.10 | 0.0085 | 84.40 |
| **EYE7** | 0.0133 | 51.36 | 0.0288 | 67.62 |

cornea were identified as the central cornea, central-peripheral, peripheral-limbus, and limbus. The mean thickness measurements following CXL treatment from central to limbus were 0.95 ± 0.08 mm, 1.03 ± 0.07 mm, 1.03 ± 0.06 mm and 1.07 ± 0.05 mm. Comparing to the control eyes, the measurements were recorded as 1.34 ± 0.09 mm (central, p = 0.003), 1.19 ± 0.05 mm (central-peripheral, p = 0.02), 1.17 ± 0.06 mm (peripheral-limbus, p = 0.02) and 1.14 ± 0.02 mm (limbus, p = 0.06) accordingly. The result demonstrated a statistically significant reduction in thickness in the central and peripheral areas (29±8% and 13±7%, respectively) which were affected by CXL.

To compare the controlled and cross-linked specimens, the material representations have been derived for corneal regions. It can be considered by the numerical parameters α and μ, in which μ is relating to the initial shear modulus and α to the non-linearity. Material behavior was compared firstly by the values of optimized material parameters α and μ which can reproduce the displacement curves of cornea apex. The inverse analysis resulted in a RMS error of 5.58 ± 1.79% (approximately 17 μm), which showed that the simulation closely matched the experimental results. The values of material parameters α and μ for all specimens using the inverse modelling procedure were generated and shown in Table 2. The average values of μ were 0.02 ± 0.012 and 0.01 ± 0.002 (p = 0.157), and the average values of α were 94.3 ± 28.9 and 65.1 ± 15.9 (p = 0.037) in CXL and PBS control, respectively. The specimens following CXL showed an increase of 45.4 ± 28.9% and 66.1 ± 110.9% in the values of α and μ. The greater values of material parameters indicated a stiffer material behavior, resulting in 27.9 ± 9.5% less displacement of cornea apex in CXL (307 ± 65 μm) than PBS (437 ± 63 μm) specimens at the IOP of 27.25 mmHg (Fig 4). The FE model provided a match of the displacements obtained from the experiment. Experimental and numerical results were examined up to an IOP of 27.5 mmHg, which demonstrated the progressive increase in corneal apex displacement as shown in Figs 4 and 5.

The corneal stress and strain curves were then extracted from the FE models and tangent modulus was calculated from the resulting stress-strain behavior. All specimens demonstrated the nonlinear behavior with an initial low tangent modulus increasing gradually under higher stress. The curve of tangent modulus (Et) versus stress (σ) for each cornea was shown in Fig 6A. The overall stiffening effect was demonstrated by the ratio of the tangent modulus larger than 1 at all stress levels (Et $_{Cross-linking}$ / Et $_{Control\ PBS}$) (Fig 6B). The average ratio values stayed between 1.4 and 1.5 throughout the inflating with IOP. Comparisons performed on tangent modulus ratios at a stress of 0.03 MPa close to the physiological level [29], a 43% ± 24% increase in tangent modulus was observed in the corneas treated with the Dresden protocol (Et $_{Cross-linking}$: 2.48 ± 0.69 vs Et $_{Control\ PBS}$: 1.73 ± 0.40, p = 0.029).

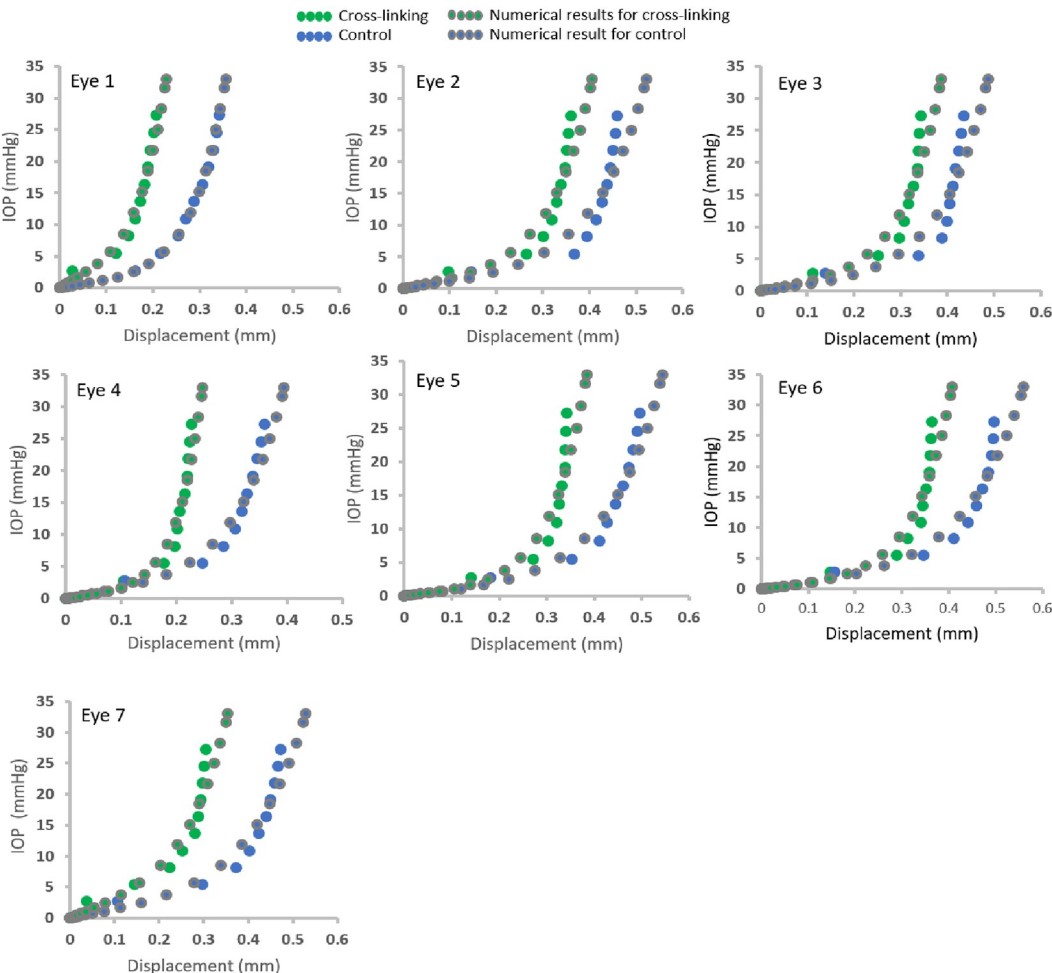

**Fig 4. IOP-displacement curves of experimental and numerical results for each individual paired eye.**

## 4. Discussion

In this study, porcine eyes were used to investigate the biomechanical behavior over the entire cornea, whole eye globes were used in generating the corneal FE model. The inverse FE method used to optimize material behavior parameters was found to provide an adequate fit between the experimental and numerical pressure-displacement behavior following CXL treatment. A significant increase was found in stiffness after CXL and a 29% reduction in corneal apical rise with increase IOP during inflation.

It has been reported that CXL significantly increased the stiffness of porcine corneas by about 42% when subjected to high pressure (300 mmHg) using inflation, but no significant difference was observed under physiological range of pressure (15 mmHg) [30]. A possible explanation could be due to the too low loading stress to observe the changes in polar distribution of fibril networks (reflecting straightening of crimp or reorientation of lamellae). The stiffening effect of cross-linked porcine cornea obtained using inflation in this study has been demonstrated to be relatively small compared to that from tensile test [6,31]. The difference in mechanical response depends on the experimental strain rate, the regions analyzed, tissues' anisotropy, and the constitutive models presented. Due to the anisotropic nature of the corneal tissue, the corneal strength is explained not only through the variations in the structure of

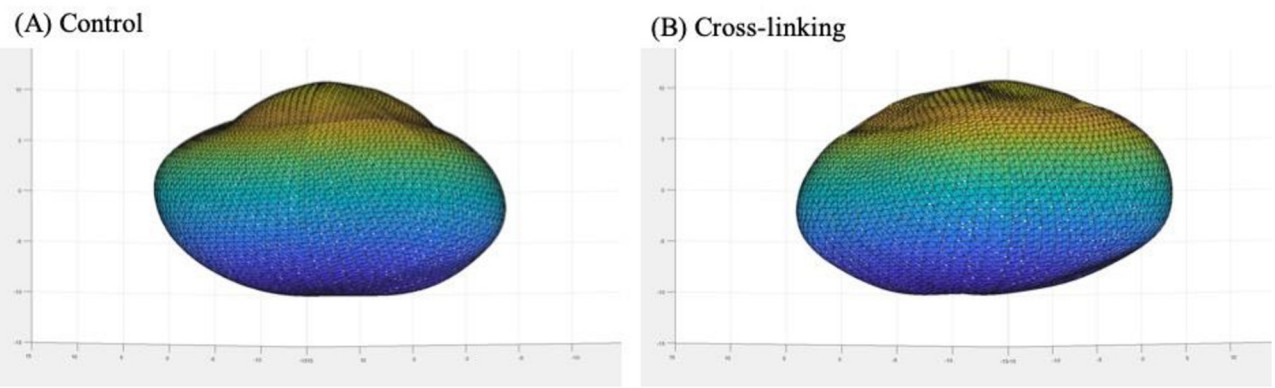

**Fig 5. Finite element model of a tested porcine cornea.** Image viewed from the side of (A) control and (B) CXL eyes.

collagen and its interactions with extracellular matrix, but also through the orientation of collagen fibrils according to the direction of the load. Collagen fibrils re-orientate themselves to the direction of the applied load, which further stiffen the tissue behavior [32]. The procedure involved in tensile test has some inherent deficiencies such as non-uniform stress distribution across the curved corneal strips, which have reduced the reliability of this method [12]. It has been reported that longitudinal fibrils are found in regions supporting tensile loads, and transverse fibrils corresponds to regions under compressive loading or loading in orthogonal direction [32,33]. Tensile and inflation testing follow different loading orientations in measurement of corneal mechanical properties. Tensile testing induces a change in collagen fibrils' alignment towards the load direction while inflation testing ensures the tissue to behave closer to the in-vivo conditions.

Additionally, the mathematical analysis of the present inflation testing was built on a number of assumptions of corneal material which was modelled as a homogenous and hyper-elastic material properties. The analysis in the current study did not take into account the variation of material stiffness between the corneal epithelium, endothelium and stroma [34], and the preferred orientation of collagen fibrils [35]. For these reasons, the analysis is expected to produce the characteristic behaviors of intact corneas which were considered as a whole in numerical simulations process.

Previous inflation study on trephination specimens which showed a rise in the apex as a function of increased IOP was very similar to the observations in porcine corneas [12]. Although the current study cannot rule out a contribution of the entire eye motion to the apical radius displacement, the behaviour which was observed in corneal apical rise was consistent in 7 pairs of eye globes. It was reported that non-cross-linked corneas did not fully return to the initial apical position after pressurized. However, cross-linked corneas tended to return to the original values both for apical position and corneal thickness, showing a more elastic behaviour than then non-cross-linked cornea [6]. The current study did not examine the hysteresis properties before and after CXL, but the results are consistent with the cross-linked cornea being stiffer.

## Limitations and future work

Although this study used a corneal model instead of a whole eye model, which was thought to be crucial to establish a biomechanical model mimicking the refractive function of a real eye [36–38]. The full parametric characterization of human corneal deformation as a function of pressure will be valuable to enhance the predictability of FE modelling of the cornea and

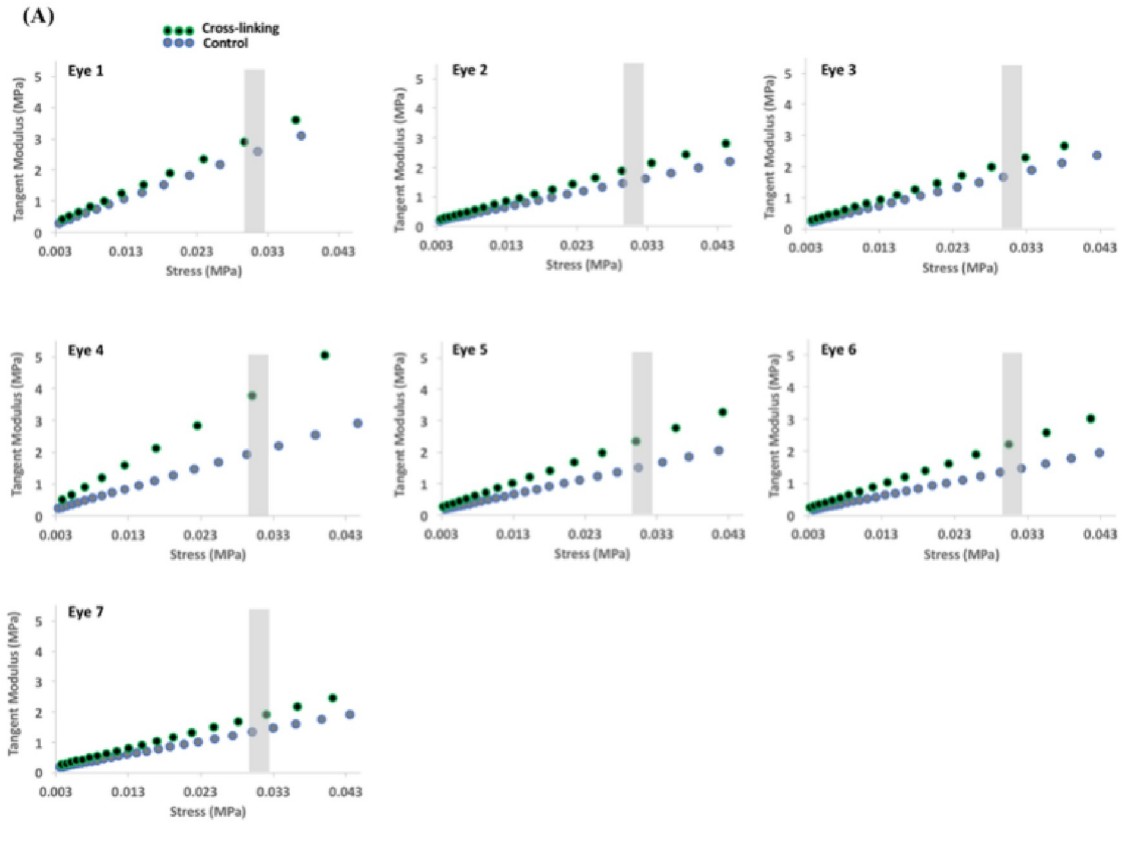

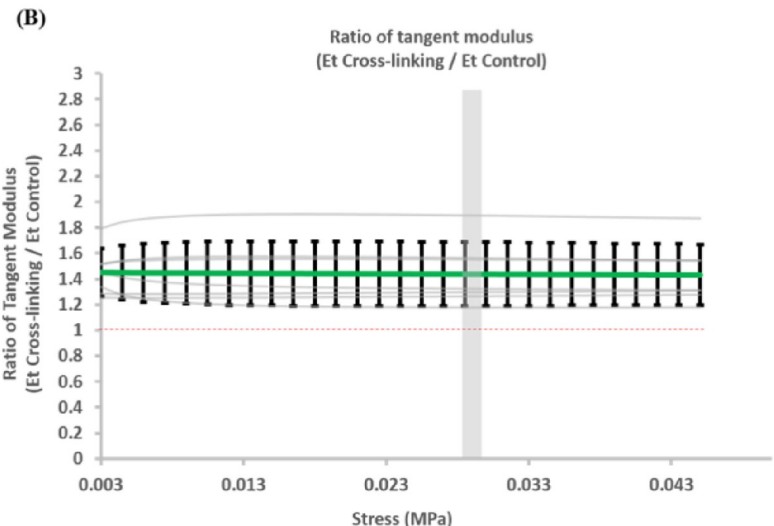

**Fig 6. The ratio of tangent modulus of paired samples.** (A) the tangent modulus vs stress behaviour of right and left eye from 7 paired eyes, (B) the ratio of tangent modulus between control and CXL (n = 7). Values from each individual pair eye tested are indicated by gray lines. Average stiffening ratio (mean ± SD) indicated by the bold line and error bars. The red dashed line represented the value of 1. Gray shaded region represents 0.03 MPa.

ultimately the predictability of the procedure. One limitation of the study was the boundary condition put on the limbus as fixed. This condition restricted the expansion of limbus in the simulation, which could underestimate the stiffness of cornea in both cross-linking treated and untreated eyes due to the externally introduced stiffness of the boundary condition. As the study chose the apex point in the calibration of material parameters, the effect of limbus expansion was considered minor. However, it was suggested to quantify the effect in our further study. Open questions such as the apparent anisotropy of the intact porcine cornea in the biomechanical response and in response to treatment are yet to be confirmed in humans and of interest in pathologic or keratoconic corneas.

Keratoconus is regarded as a degenerative disease affecting the corneal collagen networks, in which a degeneration of the collagen fibril structure and an increased propensity of fibril sliding could give rise to altered macroscopic morphology [39,40]. Therefore, using different methodologies may have profound impacts on the mechanical outcome measured, especially for the comparison of normal and diseased tissue. Further work will be needed to evaluate the effects on keratoconic corneas instead of normal corneas.

## 5. Conclusion

The current study has provided experimental data of the significant changes in corneal thickness and apical rise with increased IOP after CXL. The comparisons of stiffness and analysis of inverse FE modelling presented in this study provided important information relating to the effectiveness of CXL in biomechanical behavior across the corneas, which can be useful in applications where the prediction of the modifications of CXL protocol in corneal material stiffness is required. Although only corneal model was performed in this study, the experimental data are valuable input parameters in FE models that will allow a better understanding and increased predictability of the CXL treatment.

## Author Contributions

**Conceptualization:** Dong Zhou, Yi-Chen Li, Ahmed Elsheikh.

**Data curation:** Shao-Hsuan Chang.

**Formal analysis:** Shao-Hsuan Chang, Yi-Chen Li.

**Methodology:** Dong Zhou, Ashkan Eliasy.

**Supervision:** Ahmed Elsheikh.

**Writing – original draft:** Yi-Chen Li.

**Writing – review & editing:** Shao-Hsuan Chang, Dong Zhou, Yi-Chen Li, Ahmed Elsheikh.

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
