## [Decision Letter · Decision Letter 0]

26 Jun 2020

PONE-D-20-16381

Experimental Evaluation of Stiffening Effect Induced by UVA/Riboflavin Corneal Cross-Linking Using Intact Porcine Eye Globes

PLOS ONE

Dear Dr. Li,

Thank you for submitting your manuscript to PLOS ONE. After careful consideration, we feel that it has merit but does not fully meet PLOS ONE’s publication criteria as it currently stands. Therefore, we invite you to submit a revised version of the manuscript that addresses the points raised during the review process.

We look forward to receiving your revised manuscript.

Kind regards,

Craig Boote, PhD

Academic Editor

PLOS ONE

2. Please note that PLOS ONE has specific guidelines on software sharing (http://journals.plos.org/plosone/s/materials-and-software-sharing#loc-sharing-software) for manuscripts whose main purpose is the description of a new software or software package. In this case, new software must conform to the Open Source Definition (https://opensource.org/docs/osd) and be deposited in an open software archive. Please see http://journals.plos.org/plosone/s/materials-and-software-sharing#loc-depositing-software for more information on depositing your software.

3. We noticed you have some minor occurrence of overlapping text with previous publications, which needs to be addressed. In your revision ensure you cite all your sources (including your own works), and quote or rephrase any duplicated text outside the methods section. Further consideration is dependent on these concerns being addressed.

4. Please amend your Data availability statement to provide information of where the data used in the study can be found. We note that the data does not appear to have been provided as Supplemental information. Can other researchers obtain the dataset used?

Reviewers' comments:

Reviewer's Responses to Questions

**Comments to the Author**

1. Is the manuscript technically sound, and do the data support the conclusions?

Reviewer #1: Partly

Reviewer #2: Yes

2. Has the statistical analysis been performed appropriately and rigorously? 

Reviewer #1: Yes

Reviewer #2: Yes

3. Have the authors made all data underlying the findings in their manuscript fully available?

Reviewer #1: No

Reviewer #2: Yes

4. Is the manuscript presented in an intelligible fashion and written in standard English?

Reviewer #1: No

Reviewer #2: Yes

5. Review Comments to the Author

Reviewer #1: This study uses an inflation system to examine the mechanical properties of cornea after UVA crosslinking. It is unclear what is the actual goal of the study. There are have been numerous studies that have shown UVA crosslinking results in increases in stiffness of cornea so other than using a different measurement technique its not clear why this data is valuable. In addition, the description of the techniques and instrumentation used need to be clearer. Specific comments are listed below.

1) In the introduction the authors state "This study was also the first one to consider the stiffness of cornea as a whole following CXL." Its not clear what the authors mean here. There have been studies that have used whole corneas to evaluate CXL.

2) In section 2.1 the authors state "The internal eye components were removed through the posterior pole with a 14G needle." What components? Is it just humour or are they removing the lens, retina, etc.,

3) How large are the speckles that are applied to the eye? Do they remain attached to the same points on the eye or can the move independently? If checked, this should be stated.

4) The test rig in figure 1 should be labelled

5) There is no explanation as to how the pressure was applied, controlled or measured?

6) How was the thickness measured?

7) It is unclear how the recorded images are incorporated into the model. This should be explained better.

8) The authors should explain why they choose the Ogden model

9) In the results section the authors state "The average values of μ were 0.02 ± 0.012 and 0.01 ± 0.002 (p = 0.157), and the average values of α were 94.3 ± 28.9 and 65.1 ± 15.9 (p = 0.037) in CXL and PBS control, respectively". Should these values have units, particularly if μ represents shear modulus?

10) It is unclear how the tangent modulus was calculated. This should be explained in the methods.

Reviewer #2: This manuscript investigated the effects of cross-linking on the stiffness of cornea through inflation tests and inverse finite element modelling. The manuscript is well structured but I have several points that need to be considered.

1. The technique used to measure the corneal thickness should be introduced in the method section.

2. In the finite element model, the limbus region were restricted in the X, Y and Z direction. This limitation should also be acknowledged in the discussion section.

3. In section 2.3, last sentence. “… figure 3.6 allows …”, there is no figure 3.6 in the manuscript. I believe it should be Figure 1b.

4. In section 2.7, there is no Equation 3.2.

5. The data presented in Table 1 is inconsistent with those in Figure 3. For example, the thickness of central cornea in control and CXL groups were 1.3 and 0.95mm in Table 1, but they seems to have different values in Figure 3. Please double check those plots.

6. In the Results section (line 5-6 of first paragraph), the first group of data should be those of the control group, not the CXL group. Please check the whole manuscript thoroughly.

7. Section 2.3, ‘angler’ should be ‘angle’

6. PLOS authors have the option to publish the peer review history of their article (what does this mean?). If published, this will include your full peer review and any attached files.

Reviewer #1: No

Reviewer #2: No

---

## [Author Response · Author response to Decision Letter 0]

30 Sep 2020

Editor:

 Answer: Thanks for comments. We have prepared the manuscript according to the requirements. 

2. Please note that PLOS ONE has specific guidelines on software sharing (http://journals.plos.org/plosone/s/materials-and-software-sharing#loc-sharing-software) for manuscripts whose main purpose is the description of a new software or software package. In this case, new software must conform to the Open Source Definition (https://opensource.org/docs/osd) and be deposited in an open software archive. Please see http://journals.plos.org/plosone/s/materials-and-software-sharing#loc-depositing-software for more information on depositing your software.

Answer: Thanks for comments. We have checked the guidelines.

3. We noticed you have some minor occurrence of overlapping text with previous publications, which needs to be addressed. In your revision ensure you cite all your sources (including your own works), and quote or rephrase any duplicated text outside the methods section. Further consideration is dependent on these concerns being addressed.

 Answer: Thanks for comments. We have amended the manuscript.

4. Please amend your Data availability statement to provide information of where the data used in the study can be found. We note that the data does not appear to have been provided as Supplemental information. Can other researchers obtain the dataset used?

 Answer: Thanks for comments. We have attached the raw data as supplemental info. 

 Answer: Thanks for comments. We have linked the corresponding author’s ORCID to the account. 

Reviewer #1: This study uses an inflation system to examine the mechanical properties of cornea after UVA crosslinking. It is unclear what is the actual goal of the study. There are have been numerous studies that have shown UVA crosslinking results in increases in stiffness of cornea so other than using a different measurement technique its not clear why this data is valuable. In addition, the description of the techniques and instrumentation used need to be clearer. Specific comments are listed below.

1) In the introduction the authors state "This study was also the first one to consider the stiffness of cornea as a whole following CXL." Its not clear what the authors mean here. There have been studies that have used whole corneas to evaluate CXL.

 Answer: Thanks for comments. Yes there are many studies that have evaluated CXL using whole corneas or whole eye globe. However, most of studies focused on the tensile test, and material stiffness were measured under very high stress. While Inflation was used to evaluate the mechanical changes under different IOPs or physiological state, and the evaluation was performed by comparing the relative changes before/after CXL instead of the actual stiffness measurements. The change of stiffness was quantified through a different parameter α which comes from Ogden model to describe the elasticity of a whole cornea.

In this study, we measured the actual stiffness under physiological IOPs, which the changed amount could be much smaller than the measurements under high pressure. The method we proposed here could successfully obtain the actual stiffness and measure the differences. 

2) In section 2.1 the authors state "The internal eye components were removed through the posterior pole with a 14G needle." What components? Is it just humour or are they removing the lens, retina, etc.,

Answer: Thanks for comments. The components included viscous humour and lens, we used PBS for washing several times to remove as many components as possible. 

3) How large are the speckles that are applied to the eye? Do they remain attached to the same points on the eye or can the move independently? If checked, this should be stated.

Answer: Thanks for comments. As we have demonstrated in figure 3A, the size of the speckles ranged between 0,1(smallest)-0.5 mm(largest) in diameter. 

They remained attached to the same positions on the eyes so that we could track the displacements of the speckle at specific location/position when the eye globe inflated due to the increased IOPs. 

4) The test rig in figure 1 should be labelled

Answer: Thanks for comments. The figure has been amended in revised figures. 

5) There is no explanation as to how the pressure was applied, controlled or measured?

Answer: Thanks for comments. In this method section 2.4. A custom-built LabVIEW software was used to tightly control the pressure. The experiments started by 3 pre-conditioning cycles. The pre-conditioning cycles were to ensure the eye was sitting comfortably on the needle, and the tissue behavior was repeatable. An initial pressure of 2.5 mmHg was used to balance the external pressure applied by PBS in the pressure chamber, and was therefore considered a zero-pressure point for the inflation test. Specimens were loaded to a maximum internal load at a medium rate of 0.55 mmHg/s for each cycle. During each cycle the eye was allowed to relax for a period of 2 min. This time was obtained experimentally to allow tissue to fully recover to its relaxation state. 

LabVIEW interface. IOP and camera firing times were shown in top graph in white and yellow tick marks, respectively. 

6) How was the thickness measured?

Answer: Thanks for comments. The thickness measurement was added in revised manuscript. After the experiment was completed, the eye was removed from the test rig and dissected into anterior and posterior parts. Eight meridian profiles of discrete thickness measurements were selected as shown in Figure 2A. The thickness at each desired point on each meridian line was determined using an in-house developed Thickness Measurement Device 

(TMD) (LTA-HS, Newport, Oxfordshire, UK) which was developed by the Biomechanical Engineering group to measure the thickness of biological tissue. A vertical measurement probe was located at a height of about 30 mm above the centre point of the support. The probe moved down with a controlled velocity until it reached the surface of the tissue. By precisely knowing the original distance between the initial position of probe and the surface of support, the measured value was recorded as the thickness of the tissue.

7) It is unclear how the recorded images are incorporated into the model. This should be explained better.

Answer: Thanks for comments. To obtain the external topography from six individual profiles taken by the six cameras, the first images of the eye taken at zero pressure were selected. The location of limbus was distinguished as the border of cornea and sclera as a transition zone when building the geometry. The contour line of the eye globe was extracted by selecting the region of interest, the edge of object inside the region was automatically segmented using bespoke MATLAB codes. After the external outline was determined, the needle axis was extended to intersect the cornea. The needle (reference) was glued around the posterior pole to prevent rotation. Therefore, this intersection point was assumed as the corneal apex, as demonstrated in below figure. Therefore, the acquired results included the edge profile of each initial image of the eye, the corneal apex and limbus points. 

The internal topography was based on the external topography and the eight meridian profiles of discrete thickness measurements of cornea. The internal and external 3D topography was interpolated based on the spherical coordinates between the discrete points of measurement using bespoke MATLAB codes. In addition, a calibration ratio of mm/pixel was used to convert the obtained data into millimetre before the images were used to construct the geometry. For calibration, a steel ball with a diameter close to the size of eye (25mm) was used. Similar to above step, the region of interest around the steel ball was selected followed by segmenting the object. The best fitted circle was determined using the least square method. The mm/pixel ratio was then obtained by dividing the diameter of steel ball over the diameter of fitted circle in pixel. 

Due to the non-uniform curvature with variable thickness, the external and internal geometry of eye globe were obtained to generate the corneal model. Generally, to conduct a FE analysis in building up geometry, a given body is divided into elements which are interconnected at nodes. The nodes and elements create a network referred to as a mesh. Each element is assigned specific structural property and assembled together to give the globe response, the body is then analysed under certain boundary conditions. Therefore, the number, shape and type of elements are important factors in determining the accuracy of analysis. 

In our corneal model, the mesh of geometry was constructed with Abaqus 6.13 (Dassault Systèmes Simulia Corp., Rhode Island, USA) using bespoke software. The model was meshed using C3D15H fifteen node elements arranged in rings across the ocular surface and layers across the thickness. This element type is a second-order triangular prism with nodes at the corners and in the middle of each edge, which is capable of modelling a smooth geometric representation of variable thickness. The cornea was modelled as a homogenous, non-linear isotropic and hyper-elastic material that undergoes large deformation. Therefore, the FE was incorporated in the corneal model to capture its non-linear response under deformation. 

Segmentation results of eye globe topography 

8) The authors should explain why they choose the Ogden model

Answer: Thanks for comments. The material constitutive model chosen to describe the material behaviour of the ocular tissue during loading was Ogden model, utilised in a number of previous studies on soft tissue (Yu, Bao et al. 2013, Whitford, Joda et al. 2016). In general, Ogden model is adequate to describe the hyperelasticity of soft tissue.

9) In the results section the authors state "The average values of μ were 0.02 ± 0.012 and 0.01 ± 0.002 (p = 0.157), and the average values of α were 94.3 ± 28.9 and 65.1 ± 15.9 (p = 0.037) in CXL and PBS control, respectively". Should these values have units, particularly if μ represents shear modulus?

Answer: Thanks for comments. μ represents shear modulus and the unit is MPa, however, α has no specific unit. 

10) It is unclear how the tangent modulus was calculated. This should be explained in the methods.

Answer: Thanks for comments. This was amended in revised manuscript. The uniaxial-mode stress was calculated through obtained μ and α in Table 2, based on the previously described method (Ogden, 1972) Large Deformation Isotropic Elasticity – On the Correlation of Theory and Experiment for Incompressible Rubberlike Solids, Proceedings of the Royal Society of London. Series A, Mathematical and Physical Sciences

The tangent modulus was calculated numerically by Equation 2.3:

E =∆σ/∆ε (2.3)

Where σ is stress and ε is strain. The strain difference ∆ε in our study is 0.2%, the corresponding stress values at each strain value were presented as shown in figure 6.

Reviewer #2: This manuscript investigated the effects of cross-linking on the stiffness of cornea through inflation tests and inverse finite element modelling. The manuscript is well structured but I have several points that need to be considered.

1. The technique used to measure the corneal thickness should be introduced in the method section. 

Answer: Thanks for comments. The thickness measurement was added in revised manuscript. After the experiment was completed, the eye was removed from the test rig and dissected into anterior and posterior parts. Eight meridian profiles of discrete thickness measurements were selected as shown in Figure 2A. The thickness at each desired point on each meridian line was determined using an in-house developed Thickness Measurement Device 

(TMD) (LTA-HS, Newport, Oxfordshire, UK) which was developed by the Biomechanical Engineering group to measure the thickness of biological tissue. A vertical measurement probe was located at a height of about 30 mm above the centre point of the support. The probe moved down with a controlled velocity until it reached the surface of the tissue. By precisely knowing the original distance between the initial position of probe and the surface of support, the measured value was recorded as the thickness of the tissue.

2. In the finite element model, the limbus region were restricted in the X, Y and Z direction. This limitation should also be acknowledged in the discussion section.

Answer: Thanks for comments. Due to the non-uniform curvature with variable thickness, the external and internal geometry of eye globe were obtained to generate the corneal model. A model developed for the cornea using entire eye globe allowed appropriate realistic displacement at the limbus, the displacements at the limbus were tracked during the experiment and then introduced at the boundary of the corneal FE model. The purpose of building up corneal only model was to understand the effect of CXL treatment on corneas where the application of interest is. In addition, the approach of performing a cornel only model was to decrease the geometrical complexity and increase the efficiency of the computational calculation. Generally, to conduct a FE analysis in building up geometry, a given body is divided into elements which are interconnected at nodes. The nodes and elements create a network referred to as a mesh. Each element is assigned specific structural property and assembled together to give the globe response, the body is then analysed under certain boundary conditions. The displacements at the limbus contributed to the deformed shape of the cornea. In the present study, corneal apex was restrained against displacement in X- and Y-directions, whereas limbus was restrained in the X-, Y-, and Z-direction. Therefore, the boundary conditions were provided to avoid the model from rigid-body rotation around Z-axis. The development of numerical simulations was to generate the unique geometry of specific and generic cornea including their non-rotational symmetry and non-uniform thickness. We have added the limitation and future work in discussion.

 Limitations and future work

 Although this study used a corneal model instead of a whole eye model …. The full parametric characterization of human corneal deformation as a function of pressure will be valuable to enhance the predictability of FE modelling of the cornea and ultimately the predictability of the procedure. One limitation of the study was the boundary condition put on the limbus as fixed. This condition restricted the expansion of limbus in the simulation, which could underestimate the stiffness of cornea in both cross-linking treated and untreated eyes due to the externally introduced stiffness of the boundary condition. As the study chose the apex point in the calibration of material parameters, the effect of limbus expansion was considered minor. However, it was suggested to quantify the effect in our further study. Open questions such as the apparent anisotropy of the intact porcine cornea in the biomechanical response and in response to treatment are yet to be confirmed in humans and of interest in pathologic or keratoconic corneas.

3. In section 2.3, last sentence. “… figure 3.6 allows …”, there is no figure 3.6 in the manuscript. I believe it should be Figure 1b.

Answer: Thanks for comments. It has been amended in revised manuscript.

4. In section 2.7, there is no Equation 3.2.

Answer: Thanks for comments. It has been amended in revised manuscript. 

5. The data presented in Table 1 is inconsistent with those in Figure 3. For example, the thickness of central cornea in control and CXL groups were 1.3 and 0.95mm in Table 1, but they seems to have different values in Figure 3. Please double check those plots.

Answer: Thanks for comments. The figure was deleted and text was amended according to the value presented in Table 1. 

6. In the Results section (line 5-6 of first paragraph), the first group of data should be those of the control group, not the CXL group. Please check the whole manuscript thoroughly.

Answer: Thanks for comments. It has been amended in revised manuscript.

7. Section 2.3, ‘angler’ should be ‘angle’

Answer: Thanks for comments. It has been amended in revised manuscript.

---

## [Editor Report · Decision Letter 1]

2 Oct 2020

Experimental Evaluation of Stiffening Effect Induced by UVA/Riboflavin Corneal Cross-Linking Using Intact Porcine Eye Globes

PONE-D-20-16381R1

Dear Dr. Li,

We’re pleased to inform you that your manuscript has been judged scientifically suitable for publication and will be formally accepted for publication once it meets all outstanding technical requirements.

Kind regards,

Craig Boote, PhD

Academic Editor

PLOS ONE

Additional Editor Comments:

Thank you for your efforts in addressing the reviewers' comments in detail. I am happy to recommend publication without further referral to the reviewers in this case.

---

## [Editor Report · Acceptance letter]

22 Oct 2020

PONE-D-20-16381R1 

Experimental Evaluation of Stiffening Effect Induced by UVA/Riboflavin Corneal Cross-Linking Using Intact Porcine Eye Globes 

Dear Dr. Li:

I'm pleased to inform you that your manuscript has been deemed suitable for publication in PLOS ONE. Congratulations! Your manuscript is now with our production department. 

Kind regards, 

on behalf of

Dr Craig Boote 

Academic Editor

PLOS ONE